# Interaction of TSG101 with the PTAP Motif in Distinct Locations of Gag Determines the Incorporation of HTLV-1 Env into the Retroviral Virion

**DOI:** 10.3390/ijms242216520

**Published:** 2023-11-20

**Authors:** Yosuke Maeda, Kazuaki Monde, Hiromi Terasawa, Yuetsu Tanaka, Tomohiro Sawa

**Affiliations:** 1Department of Microbiology, Faculty of Life Sciences, Kumamoto University, Kumamoto 860-8556, Japan; monde@kumamoto-u.ac.jp (K.M.); sawat@kumamoto-u.ac.jp (T.S.); 2Department of Immunology, Graduate School of Medicine, University of the Ryukyus, Okinawa 903-0215, Japan; yuetsu@s4.dion.ne.jp

**Keywords:** HTLV-1, infectivity, retroviral Gag, PTAP, TSG101

## Abstract

Human T-cell tropic virus type 1 (HTLV-1) is known to be mainly transmitted by cell-to-cell contact due to the lower infectivity of the cell-free virion. However, the reasons why cell-free HTLV-1 infection is poor remain unknown. In this study, we found that the retrovirus pseudotyped with HTLV-1 viral envelope glycoprotein (Env) was infectious when human immunodeficiency virus type 1 (HIV-1) was used to produce the virus. We found that the incorporation of HTLV-1 Env into virus-like particles (VLPs) was low when HTLV-1 Gag was used to produce VLPs, whereas VLPs produced using HIV-1 Gag efficiently incorporated HTLV-1 Env. The production of VLPs using Gag chimeras between HTLV-1 and HIV-1 Gag and deletion mutants of HIV-1 Gag showed that the p6 domain of HIV-1 Gag was responsible for the efficient incorporation of HTLV-1 Env into the VLPs. Further mutagenic analyses of the p6 domain of HIV-1 Gag revealed that the PTAP motif in the p6 domain of HIV-1 Gag facilitates the incorporation of HTLV-1 Env into VLPs. Since the PTAP motif is known to interact with tumor susceptibility gene 101 (TSG101) during the budding process, we evaluated the effect of TSG101 knockdown on the incorporation of HTLV-1 Env into VLPs. We found that TSG101 knockdown suppressed the incorporation of HTLV-1 Env into VLPs and decreased the infectivity of cell-free HIV-1 pseudotyped with HTLV-1 Env. Our results suggest that the interaction of TSG101 with the PTAP motif of the retroviral L domain is involved not only in the budding process but also in the efficient incorporation of HTLV-1 Env into the cell-free virus.

## 1. Introduction

Human T-cell tropic virus type 1 (HTLV-1) was the first human retrovirus identified [1,2,3,4] as a cause of adult T-cell leukemia (ATL) [1,2,4] and HTLV-1-associated myelopathy/tropical spastic paraparesis (HAM/TSP) [5,6]. HTLV-1 enters target cells via specific interactions with the viral envelope glycoprotein (Env) and cellular receptor glucose transporter 1 (GLUT1). HTLV-1 Env is synthesized as a precursor of gp62 and then cleaved into a surface subunit gp46 and transmembrane gp21, which mediates viral attachment and fusion, respectively. In contrast to other retroviruses, such as human immunodeficiency virus type 1 (HIV-1), HTLV-1 is mainly transmitted by cell-to-cell contact [7] using a virological synapse [8], the formation of a biofilm-like structure [9], or a cellular conduit [10] due to the poor infectivity of cell-free HTLV-1. Thus, the effective transmission of HTLV-1 requires the clonal expansion of live-infected cells, resulting in viral persistence that eventually gives rises to ATL or chronic inflammation, such as TSP/HAM [11,12]. In contrast, cell-free pseudotyped virions of the vesicular stomatitis virus (VSV) or Moloney murine leukemia virus carrying HTLV-1 Env have been reported to be infectious [13,14,15,16], indicating that the viral core, but not the Env, of HTLV-1 may explain the poor infectivity of cell-free HTLV-1. In this study, cell-free virions carrying HIV-1 Gag-Pol pseudotyped with HTLV-1 Env conferred cell-free infectivity, and HTLV-1 Env was efficiently incorporated into virus-like particles (VLPs) carrying HIV-1 Gag, highlighting the potential role of retroviral Gag in cell-free HTLV-1 infectivity. Notably, the PTAP motif in the HIV-1 p6 domain of HIV-1 Gag interacts with tumor susceptibility gene 101 (TSG101) [17], determining the incorporation of HTLV-1 Env into VLPs and the cell-free infectivity of HIV-1 pseudotyped with HTLV-1 Env. These results indicate that the PTAP motif of retroviral Gag determines not only retroviral budding, but also the incorporation of HTLV-1 Env into cell-free virions and its infectivity.

## 2. Results

### 2.1. HTLV-1 Env Is Not Responsible for the Lower Infectivity of Cell-Free HTLV-1

We first confirmed the efficiency of HTLV-1 and HIV-1 infection through cell-free and cell-to-cell contacts using HTLV-1 and HIV-1 reporter vectors. As previously described [15], the cell-free viral infectivity of HTLV-1 was quite low compared to that of HIV-1, whereas cell-to-cell infection by HTLV-1 was successful (Figure 1A). To identify the region of HTLV-1 responsible for the lower infectivity of cell-free HTLV-1, we produced HTLV-1 and HIV-1 viruses pseudotyped with different Envs. We found that pseudotyped HTLV-1 with VSVG, but not HIV-1 and HTLV-1 Env, showed infectivity, while pseudotyped HIV-1 with HTLV-1 Env conferred infectivity (Figure 1B), indicating that HTLV-1 Gag-Pol or accessory genes may have some intrinsic defects that account for the lower infectivity of cell-free HTLV-1.

### 2.2. Inefficient Incorporation of HTLV-1 Env into VLPs Carrying HTLV-1 Gag Compared to HIV-1 Gag

Since retroviral Gag generally interacts with its Env [18,19,20,21], it is possible that HTLV-1 Gag determines the infectivity of cell-free viruses. Therefore, the role of Gag in reducing the infectivity of the cell-free HTLV-1 was evaluated. To this end, we analyzed the cell surface expression levels of HTLV-1 Env in the presence of HTLV-1 and HIV-1 Gag precursors, as the co-expression of retroviral Gag inhibited the endocytosis of retroviral Env [22]. To determine the cell surface expression levels of HTLV-1 Env, we constructed HTLV-1 and HIV-1 Gag precursor expression vectors connected to the yellow fluorescent protein, Venus [23]. The 293T cells were then transfected with HTLV-1 Env and HTLV-1- or HIV-1-Gag-Venus vectors. The cell surface expression level of HTLV-1 Env in the Gag Venus-positive cell population was analyzed by flow cytometry using an anti-HTLV-1 gp46 mAb, LAT-27 (Figure 2A) [24]. We found that the expression of HTLV-1 Env increased in the presence of HTLV-1 or HIV-1 Gag. However, no significant differences were observed between the HTLV-1 and HIV-1 Gag levels (Figure 2A). Next, we checked the fusion activity of HTLV-1 or HIV-1 Envs with target cells in the presence of HTLV-1 or HIV-1 Gag. 293T cells were transfected with HIV-1 or HTLV-1 Envs and either HTLV-1 or HIV-1 Gag connected with a FLAG-tag and co-cultured with a TZM-bl cell line carrying an integrated form of HIV-1 LTR linked to the luciferase gene. Since these Gag expression vectors also encode both rev and tat genes, the successful fusion results in the delivery of the Tat protein into TZM-bl cells [25]. The activation of the luciferase gene allows monitoring of the fusion activities of HTLV-1 and HIV-1 Env. In the presence of HIV-1 Gag, the fusion activity of HTLV-1 Env was enhanced compared to that without Gag expression, whereas HTLV-1 Gag did not enhance the cell fusion activity (Figure 2B), although HTLV-1 Gag increased the cell surface expression level of HTLV-1 Env. In contrast, the fusion activity of HIV-1 Env without Gag was comparable to that of HIV-1 Gag, whereas that of HIV-1 Env was diminished in the presence of HTLV-1 Gag (Figure 2B). Next, we examined the incorporation of HTLV-1 Env into VLPs produced by HTLV-1 or HIV-1 Gag. To produce VLPs, 293T cells were transfected with an HTLV-Env expression plasmid and either HTLV-1 or HIV-1 Gag expression vectors and recovered VLPs. Then, the incorporation of HTLV-1 Env was evaluated using LAT-27 mAb by Western blotting. Therefore, VLPs produced by HIV-1 Gag were found to efficiently incorporate HTLV-1 Env, whereas those produced by HTLV-1 Gag did not (Figure 2C).

It has been reported that cytoplasmic domain deletion or mutation affects the incorporation of HTLV-1 Env in the virions since the cytoplasmic domain has two motifs, YXXΦ and PDZ-binding, that regulate Env expression and activity via interactions with cellular proteins [26]. Therefore, it is possible that the cytoplasmic domain of HTLV-1 Env mediates its incorporation into HIV-1-Gag VLPs. However, we found that the incorporation of HTLV-1 Env with deletions or a mutation in the PDZ-binding motif of the cytoplasmic domain increased the incorporation of Envs into VLPs, as previously described (Appendix A) [26]. Taken together, the less efficient incorporation of HTLV-1 Env into HTLV-1 virions was indicative of the poor infectivity of cell-free HTLV-1.

### 2.3. Matrix Domain of HIV-1 Gag Was Not Responsible for the Efficient Incorporation of HTLV-1 Env

To further determine the domain responsible for the efficient incorporation of HTLV-1 Env, we constructed chimeras between HTLV-1 and HIV-1 Gag (Figure 3A) since the matrix domain was assumed to interact with the cytoplasmic domain of retroviral Env and contribute to the recruitment of retroviral Env [18,20,21,27]. We replaced the matrix domain of HIV-1 with HTLV-1 Gag (HTMA) and the matrix domain of HTLV-1 with HIV-1 Gag (HIMA). HTMA, but not HIMA, was found to incorporate HTLV-1 Env (Figure 3B), indicating that the matrix domain of HIV-1 Gag is not involved in the incorporation of HTLV-1 Env. A highly basic region mutant, 6A2T, in the matrix domain of HIV-1, which had lost its plasma membrane binding activity, did not produce VLPs, as previously described [28]. The deletion of the matrix domain of HIV-1 with an N-terminal Fyn (10) sequence, single myristylation, and a dual palmitoylation signal, which restores retroviral Gag targeting to the plasma membrane [29,30], produces VLPs and supports the incorporation of HTLV-1 Env (Figure 3B), confirming the role of domain(s) other than the HIV-1 matrix domain in the efficient incorporation of HTLV-1 Env. The addition of N-terminal Fyn (10) to full-length HIV-1 Gag enhanced the incorporation of HTLV-1 Env most likely due to the increased targeting of HIV-1 Gag to the plasma membrane (Figure 3B).

### 2.4. The PTAP Motif in the p6 Domain of HIV-1 Gag Is Involved in the Efficient Incorporation of HTLV-1 Env into HIV-1 VLPs

We focused on the p6 domain of HIV-1 Gag since the p6 domain in the C-terminal portion of Gag is unique to HIV-1. To evaluate the role of the HIV-1 p6 domain in the incorporation of HTLV-1 Env, we constructed PTAP or YPXL-minus p6 mutants, as well as the entire p6 deletion mutant, (Figure 4A). However, the p6 domain has been reported to be involved in viral release [17], and it is possible to decrease VLP production by these mutations. As expected, the deletion of the entire p6, PTAP, or YPXL-mutants decreased the production of VLPs, as shown in Figure 4B and Appendix A. However, we found that the overexpression of p6-deleted HIV-1 Gag and the addition of the pAdVantage vector in transfected cells increased the production of VLPs (Figure 4B) compared to that of wild-type HIV-1 Gag, most likely due to the upregulation of viral translation efficiency by adenovirus VA RNA I expression [31,32], with high amounts of HIV-1 Gag. Therefore, we were able to evaluate the incorporation of HTLV-1 Env into VLPs produced in HIV-1 p6 mutants in the presence of pAdVantage. We found that the deletion mutant of the entire p6 domain and the PTAP-minus mutant, but not the YPXL-minus mutant, did not result in the incorporation of HTLV-1 Env into the VLPs (Figure 4C), indicating the important role of the PTAP motif in the p6 domain of HIV-1 Gag in the incorporation of HTLV-1 Env into the virion.

### 2.5. VLPs Produced from HTLV-1 Gag Fused with the p6 Domain of HIV-1 Did Not Incorporate HTLV-1 Env

Given that C-terminal HIV-1 p6 Gag mediates the incorporation of HTLV-1 Env into VLPs, it is possible that HTLV-1 Gag fused to the HIV-1 p6 domain facilitates the incorporation of HTLV-1 Env into VLPs. Therefore, we constructed an expression vector for HTLV-1 Gag connected to the HIV-1 p6 domain at the C-terminus (HTLV-1-Gag-p6). Since the PTAP motif located in the C-terminal portion of the MA of HTLV-1 Gag may affect the function of HIV-1 p6, we also constructed an HTLV-1-Gag-p6 with a deleted HTLV-1 PTAP motif (HTLV-1-ΔPTAP-Gag-p6). However, these mutants did not incorporate HTLV-1 Env into the VLPs (Appendix A).

### 2.6. Role of the Interaction between the PTAP Motif and TSG101 in the Incorporation of HTLV-1 Env into HIV-1 Gag VLPs and the Infectivity of Pseudotyped HIV-1 with HTLV-1 Env

It has been reported that the p6 domain recruits the endosomal sorting complexes required for transport (ESCRT) machinery for efficient virus release [17]. For example, the PTAP motif is directly associated with TSG101, whereas the YPXL motif is associated with ALG-2-interacting protein X (ALIX). Therefore, the interaction between the PTAP motif and TSG101 may mediate the incorporation of HTLV-1 Env into HIV-1 Gag VLPs. Thus, TSG101 and ALIX were knocked down in 293T cells using shRNA lentiviral vectors (Figure 5A) [33]. Next, we produced HIV-1 Gag VLPs pseudotyped with HTLV-1 Env using these cells in the presence of pAdVantage. We found that TSG101 knockdown specifically reduced the incorporation of HTLV-1 Env into HIV-1 Gag VLPs, whereas the knockdown of ALIX or the control did not change the incorporation efficiency (Figure 5A). Conversely, the knockdown of TSG101 and ALIX did not alter the incorporation of HIV-1 Env into HIV-1 Gag VLPs (Appendix A). To confirm whether the interaction of the PTAP motif in the HIV-1 p6 domain with TSG101 determines the cell-free infectivity of pseudotyped HIV-1 with HTLV-1, the effect of TSG101 knockdown in virus-producing cells was evaluated. Our findings show that TSG101 knockdown specifically decreased the cell-free viral infectivity of pseudotyped HIV-1 with HTLV-1 Env but did not change the cell-to-cell transmission efficiency. In contrast, ALIX knockdown did not alter cell-free infectivity, whereas cell-to-cell infection was significantly enhanced (Figure 5B).

## 3. Discussion

HTLV-1 is mainly transmitted through cell-to-cell contact with HTLV-1-infected cells present in body fluids, such as breastmilk, semen, or blood, due to the low infectivity of cell-free HTLV-1 [34,35]. This poor infectivity has been partly attributed to the short half-life of HTLV-1 virions (0.6 h) at 37 °C [36]. This is likely to be due to the instability of HTLV-1 Env after it is shed from the viral particles at 37 °C [36,37]. However, previous in vitro studies have shown that cell-free viruses pseudotyped with HTLV-1 Env are infectious when viral cores are produced by other viruses, such as VSV or MLV [13,14,38,39]. These results suggest that not only Env but also HTLV-1 Gag and Gag-Pol may have intrinsic properties that lead to the poor infectivity of HTLV-1 virions. Indeed, cryoelectron tomography analysis of HTLV-1 virions has highlighted the incomplete formation of capsid cores, indicative of a defect in viral assembly [40]. In this study, we found that cell-free HIV-1, but not HTLV-1 pseudotyped with HTLV-1 Env, was infectious, suggesting that HTLV-1 Gag or Gag-Pol may partly explain the poor infectivity of HTLV-1 virions. In addition, HIV-1, but not HTLV-1 Gag, was found to enhance the incorporation of HTLV-1 Env into VLPs, indicating that the poor infectivity of cell-free HTLV-1 was partly due to the inefficient incorporation of HTLV-1 Env into cell-free virions produced by HTLV-1 Gag.

Several models have been suggested for the incorporation of retroviral Env into virions, such as passive incorporation or incorporation via direct or indirect Gag–Env interactions [41,42]. In our experiments, HTLV-1 virions pseudotyped with VSVG partially conferred infectivity (Figure 1), most likely because of the passive incorporation of VSVG into HTLV-1 virions. In contrast, the overexpression of HIV-1 or HTLV-1 Envs did not confer cell-free infectivity to HTLV-1 pseudotyped viruses. Although both Gag proteins enhanced the surface expression of HTLV-1 Env, the level of Env incorporation into VLPs was not enhanced by the co-expression of HTLV-1 Gag, in contrast to HIV-1 Gag. This suggests that mechanisms other than passive incorporation are involved in the incorporation of HTLV-1 Env into VLPs produced in HIV-1 Gag. Notably, the fusion activity of HTLV-1 Env was also enhanced by the co-expression of HIV-1 Gag, but not HTLV-1 Gag, whereas HTLV-1 Gag hampered the fusion activity of HIV-1 Env (Figure 2B). These results indicate that the interaction of retroviral Gag-Env determines both the efficient fusion and incorporation of HTLV-1 Env.

The fact that HTLV-1 Env was found to be incorporated into HIV-1 Gag VLPs suggests that MA plays a role in the incorporation of retroviral Env via direct or indirect interactions. However, the deletion mutants of MA or chimeras between HTLV-1 and HIV-1 Gag confirmed the absence of the MA domain of HIV-1 Gag for the efficient incorporation of HTLV-1 Env into HIV-1 Gag VLPs. Therefore, it is unlikely that the interaction between MA and HTLV-1 Env enhances the incorporation of HTLV-1 Env into VLPs.

We focused our analysis on the p6 domain due to its uniqueness to HIV-1. To determine the role of the p6 domain in the incorporation of HTLV-1 Env into HIV-1 Gag VLPs, it was essential to delete the p6 domain of HIV-1 Gag. However, p6 in HIV-1 Gag possesses two L-domains, the PTAP and YPXL motifs, which are involved in viral release by interacting with TSG101 and ALIX, respectively [17]. Nevertheless, we were able to rescue VLP production in the absence of the full p6 domain by overexpressing HIV-1 Gag mutants and introducing adenovirus VA RNA I (Figure 4 and Appendix A), which was initially identified as a PKR inhibitor that competed with the binding of dsRNA to PKR. Therefore, we hypothesized that higher amounts of Gag products in cells overcome the virus-releasing defect in p6 deletion mutants. However, multiple functions of VA RNA I have been reported in efficient viral replication [31,32]. Pincetic et al. showed that retroviral release was inhibited by the interferon-induced gene ISG15 [43]. As VA RNA I competes with type I interferon, it can inhibit ISG15 expression. However, further studies are needed to elucidate the mechanisms underlying the action of VA-RNA-I on viral release.

Nevertheless, Gag overexpression and the introduction of VA-RNA I in transfected cells enabled us to analyze the effect of p6 deletion mutants on the incorporation of HTLV-1 into HIV-1 Gag VLPs. We found that the deletion of p6 Gag or the PTAP minus mutation, but not the YPXL mutation, abrogated the incorporation of HTLV-1 Env into HIV-1 Gag VLPs. These findings suggest that the PTAP motif in the HIV-1 p6 domain is involved in the efficient incorporation of HTLV-1 Env into HIV-1 Gag VLPs.

Because the PTAP motif has been reported to recruit TSG101, a component of ESCRT-I complexes, for efficient viral budding from the plasma membrane, we investigated whether TSG101 is involved in the incorporation of HTLV-1 Env into HIV-1 Gag VLPs. The knockdown of TSG101, but not ALIX, in VLP-producing cells reduced HTLV-1 Env incorporation into HIV-1 Gag VLPs. Furthermore, the cell-free virus produced in TSG101, but not in ALIX knockdown cells, had lower infectivity, while infection by cell-to-cell contact was retained. These results indicate that the interaction of the PTAP motif in the p6 region with TSG101 determines the incorporation of HTLV-1 Env into HIV-1 Gag VLPs. Remarkably, HTLV-1 Gag also has a PTAP motif, which is located in the C-terminal portion of the MA domain, but plays a minor role in viral budding, in contrast to the PPXY motif, which plays an essential role [44,45,46,47,48]. Although the main function of the PTAP motif in the MA region of HTLV-1 Gag is still undetermined, these results indicate that the PTAP motif in HTLV-1 Gag does not play a role in the incorporation of HTLV-1 Env in cell-free HTLV-1. Notably, HIV-1 Gag-Pol and HTLV-1 Env-producing ALIX-knockdown 293T cells showed enhanced cell–cell infection. Although the reason for this is unclear, ALIX has been reported to bind to galectin 3, which facilitates HIV-1 budding by associating with the p6 domain of HIV-1 Gag [49]. Therefore, the inhibition of viral budding by ALIX knockdown may result in the retention of the viral assembly on the surface of infected cells, giving rise to enhanced cell-to-cell transmission.

HTLV-1 Gag fused to HIV-1 p6 did not facilitate the incorporation of HTLV-1 Env (Appendix A), indicating that a specific ternary structure of retroviral Gag is necessary for the efficient incorporation of HTLV-1 Env. It is also possible that the interaction between the PTAP motif in the p6 region of HIV-1 Gag and TSG101 induces a structural change in the immature Gag lattice, which enhances the incorporation of HTLV-1 Env. In the case of HIV-1, it has been reported that the interaction between MA and the cytoplasmic tail of Env (gp41) contributes to the incorporation of HIV-1 Env into virions [41]. Recently, X-ray crystallography and NMR structural analyses revealed that the MA lattice is arranged as a hexamer of trimers with a central hole that accommodates HIV-1 Env incorporation [46]. In contrast, even after the deletion of HIV-1, MA retained its ability to incorporate HTLV-1 Env. Therefore, the incorporation of other retroviral Envs into HIV-1 Gag VLPs is not necessary for lattice formation by HIV-1 MA. However, HTLV-1 Env was poorly incorporated into HTLV-1, most likely due to the structurally unstable core of HTLV-1 Gag [40]. Altogether, these results suggest that a well-ordered structure in the immature gag lattice, induced by the binding of HIV-1 Gag p6 to TSG101, may promote HTLV-1 Env incorporation. Conversely, infection by cell-to-cell contact is possible because the formation of a stable retroviral core structure is not necessary.

However, cell-free HTLV-1 has been reported to efficiently infect myeloid and plasmacytoid dendritic cells (DCs) via heparan sulfate proteoglycans and neuropilin-1 [50]. Thus, it is possible that small numbers of HTLV-1 Env in virions are sufficient for the efficient cell-free infection of DCs. In this study, we produced cell-free HTLV-1 using 293T cells as producer cells, whereas cell-free HTLV-1 was purified from HTLV-1-infected T cells to infect DCs [50]. Therefore, the production of cell-free HTLV-1 using T cells and the analysis of HTLV-1 Env incorporation are necessary to understand the precise mechanism underlying the incorporation of HTLV-1 Env into cell-free HTLV-1 to explain the poor infectivity of HTLV-1.

## 4. Materials and Methods

### 4.1. Cells and Culture Conditions

Human embryonic kidney 293T cells were maintained in Dulbecco’s modified Eagle’s medium (DMEM) (Sigma-Aldrich, St. Louis, MO, USA) supplemented with 10% fetal bovine serum (FBS) (Gibco BRL, Carlsbad, CA, USA). A glioma cell line expressing human CD4 (NP-2/CD4) and its derivative, NP-2/CD4/CCR5/CXCR4 [51,52,53], was maintained in Eagle’s minimum essential medium (Sigma-Aldrich) supplemented with 10% FBS. The TZM-bl cell line [25] was obtained from the NIH AIDS Research and Reference Reagent Program, Division of AIDS, National Institute of Allergy and Infectious Diseases, and maintained in DMEM supplemented with 10% FBS.

### 4.2. Plasmids

The HTLV-1 Env expression vector pcDNA-1E-RRE was constructed as previously described [52]. HIV-1 and HTLV-1 intron Gaussia luciferase reporters (pUCHR-inGLuc and pCRU5-HT-inGLuc) and packaging plasmids for HTLV-1 and HIV-1 (pCMVPA-HIV and pCMVHT1MDEnv) for cell-free and cell-to-cell infection [15] were kindly provided by Dr. David Derse and Gisela Heidecker (National Cancer Institute, Bethesda, MD, USA). The plasmid-enhancing translation initiation vector pAdVantage was purchased from Promega (Madison, WI, USA). Expression vectors for HIV-1 Gag, pCRVI/HIV-1/Gag-FLAG-tag, pCRVI/HIV-1/Gag-HA-tag, and pCRVI/HIV-1/Gag-Venus were constructed as previously described [53]. HIV-1 Gag mutants containing N-terminal Fyn and 6A2T mutations in the highly basic region of HIV-1 MA have been previously described [30]. Lentiviral vectors for the constitutive expression of shRNAs targeting TSG101 and ALIX [33] were kindly provided by Dr. Yasuo Ariumi (Nagasaki University, Nagasaki, Japan).

### 4.3. Construction of Expression Vectors

Expression vectors for HTLV-1-Gag-FLAG and -Venus were constructed from pCRVI/HIV-1/Gag-FLAG or pCRVI/HIV-1/Gag-Venus by replacing the HIV-1 gag sequences with HTLV-1 gag sequences amplified by PCR using the pMT-2 HTLV-1 plasmid [52] as the template. Chimeric Gag expression vectors between HTLV-1 and HIV-1 were constructed from pCRVI/HTLV-1/Gag-FLAG and pCRVI/HIV-1/Gag-FLAG, respectively, using standard molecular techniques. HTLV-1 Gag mutants containing the Fyn N-terminus were constructed in a similar manner. A Gag expression vector with an entire deletion of p6 domain was constructed from pCRVI/HIV-1/Gag HA. Gag expression vectors containing the PTAP or YPXL mutants were constructed from pCRVI/HIV-1/Gag-HA using site-directed mutagenesis.

### 4.4. Establishment of 293T Cells Constitutively Knocking down TSG101 or ALIX

The lentiviral vector particles were produced via the transient transfection of lentiviral vectors expressing shRNAs targeting TSG101 and ALIX with packaging and VSVG-expressing plasmids, as previously described [33]. Lentivirus-transduced 293T cells were then selected in a culture medium containing 1 μg/mL puromycin.

### 4.5. Cell-Free or Cell-To-Cell Infection Assay

For cell-free or cell-to-cell infection assays for HTLV-1 and HIV-1, an intron Gaussia reporter system (inGLuc) was used, as previously described [15]. Briefly, 293T cells were transfected with HTLV-1 or HIV-1 inGLuc reporter vectors, packaging constructs, and Env expression plasmids for HIV-1 and HTLV-1. After 24 h of transfection, the culture supernatant was recovered, and the same volume was used for the cell-free infection of NP-2/CD4/CCR5/CXCR4 cells. For cell-to-cell infection, the transfected cells were washed and mixed with an equal number of NP-2/CD4/CXCR4/CCR5. After 48 h of cell-free infection or co-culture, the Gaussia luciferase activity of the culture supernatant was measured using a BioLuc Gaussia Luciferase Assay Kit (New England BioLabs, Ipswich, MA, USA) and luminometer (Berthold Technology, Bad Wildbad, Germany). The infectivity of the cell-free pseudotyped virus was normalized to the luciferase activity per 1 ng of p24 Ag of HTLV-1 and HIV-1.

### 4.6. p24 Antigen ELISA of HIV-1 and HTLV-1

The p24 antigen concentration of pseudotyped HIV-1 was determined using p24Ag ELISA (Zeptometrix, Buffalo, NY, USA) according to the manufacturer’s instructions. The concentration of p24 antigen in the pseudotyped HTLV-1 was determined by ELISA using two monoclonal antibodies against p24 Ag of HTLV-1.

### 4.7. Cell Fusion Assay

For the cell-to-cell fusion assay, 293T cells in a six-well plate were transfected with 3.5 µg of pCRVI/HTLV-1/Gag-FLAG or pCRVI/HIV-1/Gag-FLAG, 1.5 µg of pcDNA-1E-RRE or HIV-1 Env expression plasmid (pCXN-NLenv [52]), and 0.5 µg of pAdVantage (Promega) using a Profection kit (Promega) or Lipofectamine 2000 (Thermo Fisher). Transfected 293T cells were recovered at 6 h post-transfection and co-cultured with TZM-bl cells. Luciferase activity was measured 30 h post-transfection using a luminometer (Berthold Technology), as previously described [52].

### 4.8. Production of VLPs

To produce VLPs, 293T cells were transfected with either 3.5 µg of pCRVI/HTLV-1/Gag-FLAG, or pCRVI/HIV-1/Gag-FLAG, 1.5 µg of pcDNA-1E-RRE, and 0.5 µg of pAdVantage (Promega). The culture supernatant of the transfected cells was recovered, filtered, and used for VLP analysis. To analyze the HIV-1 Gag p6 mutants, the pCRVI/HIV-1/Gag-HA plasmid was used for transfection instead of pCRVI/HIV-1/Gag-FLAG.

### 4.9. Flow Cytometry

The 293T cells were transfected with the HTLV-1 Env expression plasmid (pcDNA-1E-RRE) and either pCRVI/HTLV-1/Gag-Venus or pCRVI/HIV-1/Gag-Venus, as described above. The cells were recovered 24 h post-transfection using a cell dissociation solution (Sigma-Aldrich), stained with the anti-gp46 rat mAb LAT-27 [24]. The cells were then further stained with APC-conjugated anti-rat IgG (Bioligands, San Diego, CA, USA) and fixed with 4% paraformaldehyde for 15 min. Next, the cells were acquired using a FACSCalibur fluorescent-activated cell sorter (BD Biosciences, San Jose, CA, USA) and analyzed using FlowJo software version 10.9.0 (FlowJo, LLC, Ashland, OR, USA).

### 4.10. Western Blotting

Transfected 293T cells were solubilized in 1% Brij O10 (Sigma-Aldrich) lysis buffer (1% Briji O10, 20 mM Tris-Cl pH 8.0, and 150 mM NaCl) containing a protease inhibitor cocktail (Nacalai Tesque, Kyoto, Japan). For the analysis of VLPs by Western blotting, the supernatant of transfected 293T cells was centrifuged, and the resulting pellets were solubilized in 2× sample loading buffer. Cell lysates or solubilized VLPs were separated by SDS-PAGE and blotted onto polyvinylidene fluoride membranes (Immobilon-P; Millipore, Billerica, MA, USA). The membranes were then incubated with anti-gp46 rat mAb (LAT-27), anti-gp120 polyclonal sheep antibody (Aalto Bio Reagents, Dublin, Ireland), anti-FLAG mouse mAb (Wako, Osaka, Japan), anti-HA mouse mAb (Wako), anti-cyclophilin A (CypA) rabbit pAb (Enzo Life Sciences, Farmingdale, NY, USA), or anti-β-actin mouse mAb (Sigma), followed by staining with horseradish peroxidase-conjugated anti-mouse (Jackson ImmunoResearch, West Grove, PA, USA), anti-rat (Jackson ImmunoResearch), or anti-rabbit (Jackson ImmunoResearch) IgGs. Signals were detected using Chemi-Lumi One (Nacalai Tesque). The images were then acquired using ChemiDoc Touch (Bio-Rad, Hercules, CA, USA) and analyzed using Image Lab Software version 6.1.010.9.0 (Bio-Rad).

## 5. Conclusions

In this study, we investigated the reasons for the poor infectivity of cell-free HTLV-1. Our results indicated that HTLV-1 Env was efficiently incorporated into VLPs produced by HIV-1 but not HTLV-1 Gag, highlighting the role of HTLV-1 Gag in the lower infectivity of cell-free HTLV-1. We further verified that the interaction of the PTAP motif located in the p6 domain of HIV-1 Gag with TSG101 was responsible for the incorporation of HTLV-1 Env into HIV-1 Gag VLPs and enhanced the cell-free infectivity of pseudotyped HIV-1 with HTLV-1 Env. Overall, these findings indicate that the poor infectivity of HTLV-1 can be partly attributed to the inefficient incorporation of HTLV-1 Env into cell-free HTLV-1.

## Figures and Tables

**Figure 1 ijms-24-16520-f001:**
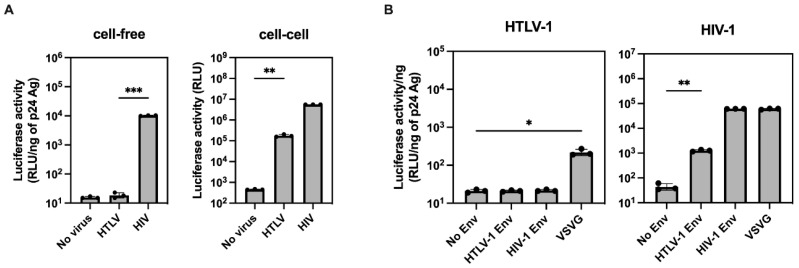
Cell-free and cell-to-cell infection by HIV-1 and HTLV-1, and pseudotyped virus infection with different Envs. (**A**) The same volume of culture supernatants from 293T cells transfected with HTLV-1-inGLuc and HIV-1-inGLuc and their respective packaging vectors and Env expression plasmid were used for the cell-free infection of NP-2/CD4/CCR5/CXCR4 cells. Transfected cells were added to NP-2/CD4/CCR5/CXCR4 cells for cell-to-cell infection. The Gaussia luciferase activity of cell-free and cell-to-cell infection was measured after 48 h of infection. The cell-free virus infectivity was normalized by luciferase activity per 1 ng p24 Ag of HTLV-1 and HIV-1, respectively. The untransfected 293T cells were used as the no-virus control. The column and bar indicate the mean and standard deviation in triplicate experiments, respectively (** *p* < 0.01, *** *p* < 0.001 using unpaired *t*-test). (**B**) Pseudotyped virus produced from 293T cells transfected with HTLV-1-inGLuc or HIV-1-inGLuc vectors; different Envs were used for the cell-free infection of NP-2/CD4/CCR5/CXCR4 cells. The Gaussia luciferase activities of culture supernatant from infected cells were measured after 48 h infection. The pseudotyped virus infectivity was normalized by luciferase activity per 1 ng p24 Ag of HTLV-1 and HIV-1, respectively. The column and bar indicate the mean and standard deviation in triplicate experiments, respectively (* *p* < 0.05, ** *p* < 0.01 using unpaired *t*-test).

**Figure 2 ijms-24-16520-f002:**
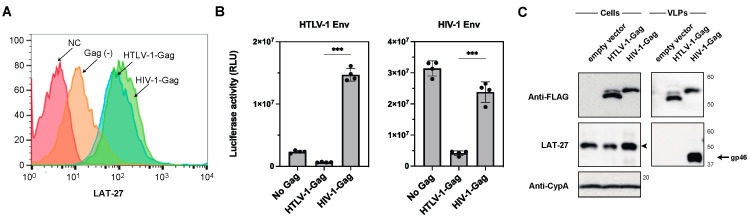
HTLV-1 Env expression level and fusion activity in the absence or presence of retroviral Gag, and efficiency of HLTV-1 Env incorporation into VLPs produced by retroviral Gag. (**A**) The HTLV-1 Env expression plasmid was transfected into 293T cells with control Venus vector (orange), HTLV-1-Gag-Venus (blue), and HIV-1-Gag-Venus (green). After 24 h of transfection, the cells were recovered and stained with a monoclonal antibody against gp46, LAT-27. Venus-positive cell populations were analyzed by flow cytometry. Red indicates a lack of antibody staining (control). (**B**) The HTLV-1 Env expression plasmid was transfected into 293T cells with control FLAG vector, HTLV-1-Gag-FLAG, and HIV-1-Gag-FLAG plasmids. After 24 h of transfection, transfected cells were recovered and added to TZM-bl cells. The luciferase activities of mixed cells were measured after 24 h of co-culture. The column and bar indicate the mean and standard deviation in quadruplicate experiments, respectively (*** *p* < 0.001 using unpaired *t*-test). (**C**) Cell lysates and VLPs produced from 293T cells transfected with control vector, HTLV-1-Gag-FLAG, or HIV-1-Gag-FLAG combined with HTLV-1 Env were analyzed by Western blotting using anti-FLAG, LAT-27, and anti-CypA antibodies. The arrowhead shows the ~53-kDa size of the Env protein. The positions of the molecular mass marker (kDa) are indicated on the right.

**Figure 3 ijms-24-16520-f003:**
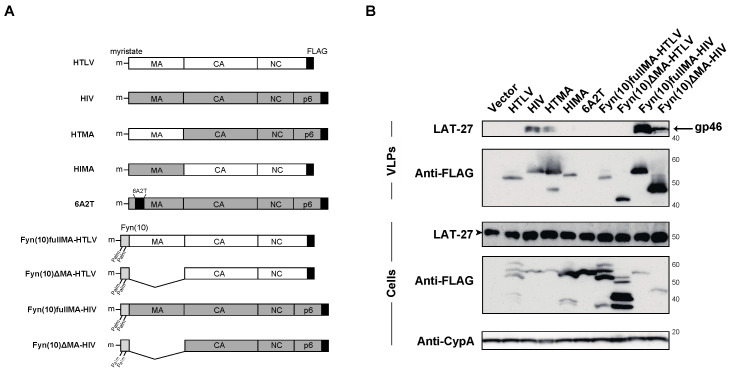
Efficiency of HTLV-1 Env incorporation into VLPs produced in chimeras between HTLV-1 and HIV-1 Gag and mutants in HTLV-1 and HIV-1 Gag. (**A**) Schematic representation of chimeras between HTLV-1 and HIV-1 Gag and mutants in HTLV-1 and HIV-1 Gag. (**B**) The cell lysates and VLPs produced from 293T cells transfected with HTLV-1 Env and chimeras between HTLV-1 and HIV-1 Gag or mutants in HTLV-1 and HIV-1 Gag were analyzed by Western blotting using anti-FLAG, LAT-27, and anti-CypA antibodies. The arrowhead shows the ~53-kDa size of the Env protein. The positions of the molecular mass marker (kDa) are indicated on the right.

**Figure 4 ijms-24-16520-f004:**
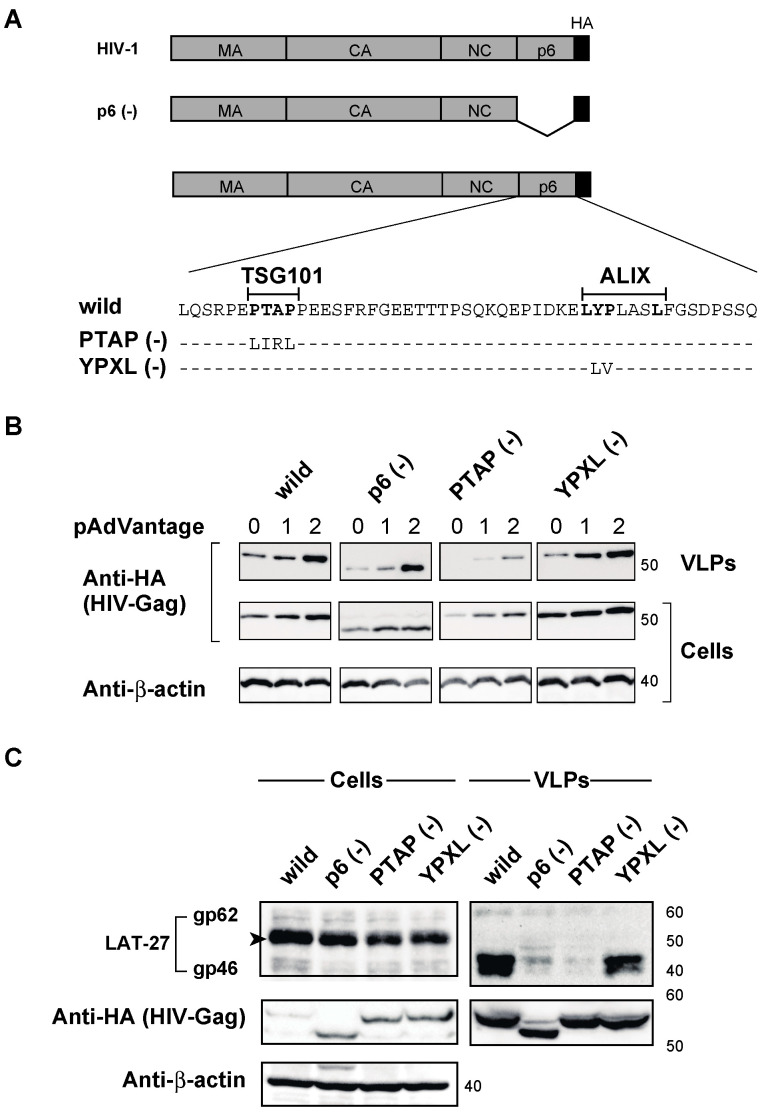
Efficiency of HTLV-1 Env incorporation into VLPs produced in HIV-1 p6 Gag mutants. (**A**) Schematic representation of the deletions and mutations in the p6 domain of HIV-1-Gag. Amino acid sequences of the wild-type p6 domain are shown with the PTAP and YPXL motifs. A dash denotes amino acid identity. (**B**) Cell lysates and VLPs produced from 293T cells transfected with 1 μg of p6 Gag deletion and mutants with different amounts of pAdVantage (0, 1, and 2 μg) were analyzed by Western blotting using anti-HA and anti-β-actin antibodies. (**C**) Cell lysates and VLPs produced from 293T cells transfected with 1.5 μg of HTLV-1 Env and 3.5 μg of p6 deletion and mutants of HIV-1-Gag-HA were analyzed by Western blotting using anti-HA, LAT-27, and anti-β-actin antibodies. The arrowhead shows the ~53-kDa size of the Env protein. The positions of the molecular mass marker (kDa) are indicated on the right.

**Figure 5 ijms-24-16520-f005:**
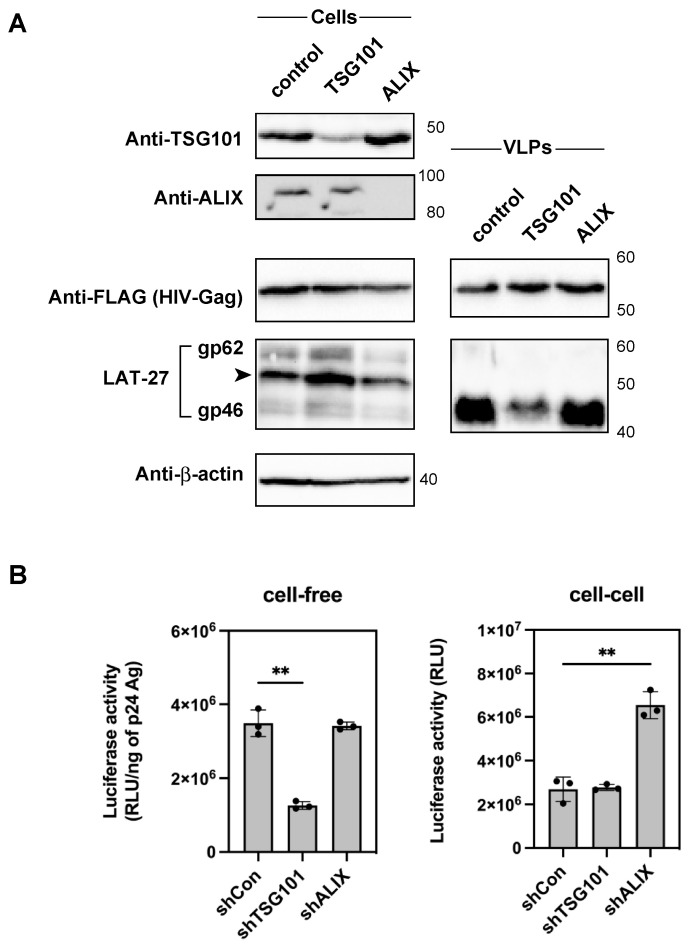
Incorporation of HTLV-1 Env in HIV-1-Gag VLPs and cell-free and cell-to-cell infectivity of HIV-1 Gag pseudotyped with HTLV-1 Env produced in 293T cells with TSG101 or ALIX knockdown. (**A**) Expression levels of TSG101 and ALIX in 293T cells transduced with shRNA targeting control, TSG101, and ALIX were analyzed by Western blotting using anti-TSG101 and anti-ALIX antibodies. The knockdown 293T cells were transfected with HIV-Gag-FLAG and HLTV-1 Env, and the resulting cell lysates and VLPs were recovered after 48 h of transfection and analyzed by Western blotting using anti-FLAG, LAT-27, and anti-β-actin antibodies. The arrowhead shows the ~53-kDa size of the Env protein. The positions of the molecular mass marker (kDa) are indicated on the right. (**B**) TSG101 or ALIX knockdown 293T cells were used to produce HIV-1 in GLuc with HIV-1 packaging vectors and the HTLV-1 Env expression plasmid. The same volume of culture supernatant of transfected cells was used for the cell-free infection of NP-2/CD4/CCR5/CXCR4 cells. Transfected cells were added to NP-2/CD4/CCR5/CXCR4 cells for cell-to-cell infection. The Gaussia luciferase activity of the resulting culture supernatant was measured after 48 h of culturing for cell-free and cell-to-cell infection. The cell-free virus infectivity was normalized by luciferase activity per 1 ng p24 Ag of HIV-1. The 293T cells transduced with a control shRNA vector (shCon) were used as a control. The column and bar show the mean and standard deviation in triplicate experiments, respectively (** *p* < 0.01 using unpaired *t*-test).

## Data Availability

All data associated with this study are presented in the paper or Appendix A. All the raw data are available upon request.

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
