# Peer review of "Interaction of TSG101 with the PTAP Motif in Distinct Locations of Gag Determines the Incorporation of HTLV-1 Env into the Retroviral Virion"

_ijms, 2023, doi:10.3390/ijms242216520_

Round 1

Reviewer 1 Report

Comments and Suggestions for Authors

This paper by Maeda et al seeks to address the mechanisms underlying the observed lack of cell-free infectivity exhibited by HTLV-1. The author's use a pseudotyping strategy to separately assess the role HTLV-1 Env and Gag in the HTLV-1 cell free infectivity. The authors find that HIV-1 Gag VLPs are much more efficient in incorporating HTLV-1 Env than HTLV-1 Gag VLPs. This property is dependent on the p6 domain of HIV-1, which is not present in HTLV-1 Gag. Furthermore, the ability of HIV-1 Gag to incorporate HTLV-1 Env is partially dependent on the interaction between the PTAP motif of p6 and TSG101. While this is interesting as an individual result, there are technical problems with the study that must be addressed. Furthermore, I am not sure what the punchline is for the work as I find myself left with more questions than answers. Specific comments below. 

Major issues

Figure 4 is missing from the paper! I had to go to the "original images of the blots and gels" to find the data that was being referenced. Additionally, the images here were not originals, just reprints of exactly what was already in the figures. My understanding is that the point of seeing the original images is transparency regarding the cropping and arranging of the images for the figures. There is no point in publishing these as "original" 

As far as I can tell, there is no normalization of viral inputs for the infectivity assays. This is a huge problem especially when virus production would be expected to be affected by p6 deletions/mutations. Also, the differences in infectivity described in fig 5b are quite small and could have more to do with differences in viral input as opposed to actual infectivity. It also may explain the unknown reason behind VLPs produced from ALIX KD cells being more infectious than those produced from WT cells. Regardless, the normalization method or volume of supernatant used in these experiments must be published in the methods. 

I am quite skeptical of the p6 deletion data. The first set of panels in fig 4 suggests that even upon co-transfection with pAdVantage, the p6 del and p6 mutants show considerably lower VLP production than WT. Yet in the next figure, the production is the same for all mutants and WT. How is pAdVantage restoring the p6del-induced defect on VLP production?

Is part of the takeaway from fig 2 that HTLV-1 Gag is essentially unable to incorporate its own Env into VLPs? Despite increasing its cell surface expression? This doesn't seem likely to me and needs to be put in context with other studies. Is this commonly seen with HTLV-1? 

Fig 2B should also be performed with HIV-1 env. If HTLV-1 Gag also decreases the fusogenicity of HIV-1 env, the conclusion would be more convincing. 

Similarly, I think that fusing the HIV-1 p6 domain to HTLV Gag to see if this increases HTLV-1 env incorporation would significantly strengthen the paper. 

Minor points

The authors about pAdVantage in the results and say they don't know why it works, but then explain it in detail in the discussion without actually saying this is pAdVantage. 

Add a key for the key in fig 2a

Please MW markers to the blots

What is 6A2T (Fig 3B)? I didn't see it mentioned in the text. 

Comments on the Quality of English Language

The paper will require some moderate editing to make things more concise. The discussion should probably be split into smaller sections to make it more digestible. 

Author Response

Response to Reviewer 1 Comments

1. Summary

2. Response and Revisions

We thank the reviewer’s comments. These comments were helpful, and gave us a better perspective of our work. The point-by-point comments were as bellows.

3. Point-by-point response to Comments and Suggestions for Authors

Comments 1: Figure 4 is missing from the paper! I had to go to the "original images of the blots and gels" to find the data that was being referenced. Additionally, the images here were not originals, just reprints of exactly what was already in the figures. My understanding is that the point of seeing the original images is transparency regarding the cropping and arranging of the images for the figures. There is no point in publishing these as "original" 

Response 1: First of all, we have to apologize the missing of Figure 4 in the manuscript. We added the Figure 4 in the revised manuscript and added the original images of the Western blot in the supplement.

Comments 2: As far as I can tell, there is no normalization of viral inputs for the infectivity assays. This is a huge problem especially when virus production would be expected to be affected by p6 deletions/mutations. Also, the differences in infectivity described in fig 5b are quite small and could have more to do with differences in viral input as opposed to actual infectivity.

Response 2: We appreciate the reviewer’s comments regarding the normalization of viral input. Actually, we determined the virus input of HTLV-1 and HIV-1 using ELISA of p24 antigen of HTLV-1 and HIV-1, respectively, and measured the luciferase activity per 1 ng of p24 Ag for the normalization of the virus input. Therefore, we changed the Y axis title to “luciferase activity (RLU/ng of p24 Ag)” in Figure 1 and Figure 5B for cell-free infection. Normalization method of virus input in cell-free infection assay were incorporated in the figure legends (Line 71-72, 77-78, 238-239) and materials and methods (Lines 394-396). Methods for ELISA of HTLV-1 and HIV-1 p24 Ag were also added in materials and methods at Lines 397-401.

Comments 3-1: It also may explain the unknown reason behind VLPs produced from ALIX KD cells being more infectious than those produced from WT cells. Regardless, the normalization method or volume of supernatant used in these experiments must be published in the methods. 

Response 3-1: As the reviewer suggested, difference between cell-free virus infectivity produced in TSG101-knock down and control cells was small. However, we found that the comparable levels of p24 antigen were produced in control, TSG101- and ALIX-knock-down cells, and found the statistical significance in the same virus input normalized by p24 Ag as we mentioned earlier.

The reviewer also asked the reason why the ALIX-knockdown cells enhanced the cell-cell but not cell-free infection compared to control cells. It is quite difficult to answer this question at the current status, but it has been reported that ALIX binds to galectin 3, which facilitates the HIV-1 budding via association with HIV-1 Gag p6 {Wang, 2014 #99}. Therefore, inhibition of viral budding by ALIX-knockdown is possible to cause the retention of viral assembly in the surface of infected cells, resulting in the enhancement of the cell-to-cell transmission. These were incorporated in the discussion at Lines 314-318.

Comments 3-2: I am quite skeptical of the p6 deletion data. The first set of panels in fig 4 suggests that even upon co-transfection with pAdVantage, the p6 del and p6 mutants show considerably lower VLP production than WT. Yet in the next figure, the production is the same for all mutants and WT. How is pAdVantage restoring the p6del-induced defect on VLP production?

Response 3-2: We appreciate the reviewer’s comments regarding the effect of pAdVantage. One of the reasons why pAdVantage restores the viral releasing inefficiency in p6 deletion mutants is upregulation of translation in infected cells by competing the binding of dsRNA to PKR which activates the phosphorylation of eIF2α. We further used higher amount of HIV-1-Gag-FLAG contrast (3.5μg) in Figure 4C while we used the lower amount of HIV-1 Gag construct (1μg) in Figure 4B to see the effect of p6 deletion mutants. The figure showing the effect of pAdVantage and high amount of HIV-1 Gag p6 deletion mutant were added in the revised manuscript. These effects may be enough to compensate the expression levels of VLPs in p6 deletion mutants of HIV-1 Gag constructs though the additional mechanism of pAdVantage was unknown. We think further studies will be necessary to elucidate the additional effect of VA RNA I in the viral budding process. These were incorporated in the discussion at Lines 284-293.

Comments 4: Is part of the takeaway from fig 2 that HTLV-1 Gag is essentially unable to incorporate its own Env into VLPs? Despite increasing its cell surface expression? This doesn't seem likely to me and needs to be put in context with other studies. Is this commonly seen with HTLV-1? 

Response 4: We appreciate the reviewer’s comments regarding the inefficient incorporation of HTLV-1 Env in its own VLPs. As we mentioned in the discussion, it is possible to incorporate the HTLV-1 Env into in the HTLV-1 virions when certain cell-types were used such as dendritic cells. However, most of previous studied have shown the cell-free infectivity of HTLV-1 was lost by shedding of Env due to the unstable nature of HTLV-1 Env. Our study added the new insight that HTLV-1 Gag also affects the incorporation efficiency of HTLV-1 Env into the virions. These were incorporated in discussion part at Lines 336-344.

Comments 5-1: Fig 2B should also be performed with HIV-1 env. If HTLV-1 Gag also decreases the fusogenicity of HIV-1 env, the conclusion would be more convincing. 

Response 5-1: We appreciate the reviewer’s suggestion. We conducted the experiments according to the reviewer’s suggestion to check the fusion activity of HIV-1 Env in the presence of HTLV-1 and HIV-1 Gag. Since HIV-1 Env has higher fusogenicity compared to HTLV-1 Env, we found HIV-1 Env alone had maximum activity while HTLV-1 Gag expression reduced the HIV-1 Env fusion activity. These findings also support our idea that HTLV-1 Gag may hamper the several functions of Env. We therefore renewed the Figure 2B adding the fusion activity of HIV-1 Env in the presence of HTLV-1 and HIV-1 Gag. These results were incorporated in the revised manuscript accordingly.

Comments 5-1: Similarly, I think that fusing the HIV-1 p6 domain to HTLV Gag to see if this increases HTLV-1 env incorporation would significantly strengthen.

Response 5-2: We appreciate the reviewer’s suggestion. We actually constructed the HTLV-1 Gag fused with p6 domain of HIV-1 Gag in the C-terminal portion, and checked the incorporation efficiency of HTLV-1 Env. However, HTLV-1 Env was not incorporated into VLPs produced in the chimeric Gag. This indicated the efficient incorporation of Env was not simply explained by the addition of L domain in the retroviral Gag at the C-terminal portion. We think that a ternary structure of retroviral Gag with L domain would be necessary for the incorporation of HTLV-1 Env. These results and discussions were incorporated in Results at Lines 195-203, and Discussions at Lines 319-335.

Minor points

The authors about pAdVantage in the results and say they don't know why it works, but then explain it in detail in the discussion without actually saying this is pAdVantage. 

We thank reviewer’s comment. As we mentioned earlier, we discussed these reasons in discussion part in revised manuscript in Discussions at Lines284-293.

Add a key for the key in fig 2a

We gave the name of each histogram in Figure 2A.

Please MW markers to the blots

We added the MW markers to the blots.

What is 6A2T (Fig 3B)? I didn't see it mentioned in the text. 

We added the sentences explaining the 6A2T Gag mutant which cannot translocate to the plasma membrane, hence does not release VLPs in the text with the reference at Lines 147-149.

The paper will require some moderate editing to make things more concise. The discussion should probably be split into smaller sections to make it more digestible. 

We split the discussion into the small sections. The revised manuscript has been further edited by another native speaker.

Reviewer 2 Report

Comments and Suggestions for Authors

Maeda et al. investigated why cell-free HTLV-1 virus has poor infectivity. They found that HTLV-1 Env can be efficiently incorporated into the VLPs with HIV-1 Gag but not HTLV-1 gag, and the pseudotyped virus with HIV-1 Gag has increased cell-free infectivity. The PTAP motif in the p6 domain of HIV-1 Gag interacts with TSG101 to incorporate HTLV-1 Env into the VLP. The experiments were well designed, but the explanation of the methods and results can be improved. These are also editorial errors (repeated 2.4 paragraphs and missing figure 4 in the manuscript pdf) which dampen the quality of the submission (not sure if these errors are caused by the journal or the authors’ mistakes).

Comments:

1. Line 107: Add (Fig. 2C) after “did not.”

2. Fig 2C left panel: Explain why the anti-flag band in HIV-1 Gag cells in cells is lower than HTLV-1-Gag.

3.     Line 128-130: correct the grammatical error of “Cell lysates and VLPs produced from 293T cells transfected with HTLV-1 Env and control vector, HTLV-1-Gag-FLAG, or HIV-1-Gag-FLAG were analyzed by Western blot using anti-FLAG, LAT-27, and anti-CypA antibodies.”

4.      Line 142-147: Explain why N-terminal Fyn was added (containing myristylation and palmitoylation sites) so it can target gag to the membrane. It is mentioned in the discussion, but it should be explained here as well.

5.      Line 142-147: Explain why 6A2T mutation is included in the screen. What is special about this region?

6.      Line 161: the whole figure 4 is missing in the manuscript pdf, and only two figures are included in the original images of blots/gels so one sub-figure is totally missing.

7.      Line 161-163: please add references of p6 involvement in viral release, and the references of the association of L-domain with VLPs production.

8.      Line 172-188: this part is a duplication. Not sure if this is journal or authors’ error.

9.      Line 200: (Fig. 5B) should be (Fig. 5A)

10.   Figure S2: please improve the quality of this blot. Explain why gp120 band in control cells is lower, and why actin bands are not equal among the 3 lysates.

11.   Line 208: Add “(Fig. 5B)” at the end of the sentence.

12.   Line 263-265: Correct the grammatical error of “p6 domains possess two L-domains, PTAP and YPXL motifs, and are involved in viral release by interaction with TSG101 and ALIX, respectively.”

13.   Line 265-275: The discussion of VA RNA I is confusing. Where is it located? What is the relationship between VA RNA I with p6, PTAP or YPXL?

14.   Line 375: Explain the importance of pcDNA-1E-RRE vector.

Comments on the Quality of English Language

Please correct some errors 

Author Response

Response to reviewer 2 comments

1. Summary

2. Response and Revisions

We thank the reviewer’s comments. These comments were helpful, and gave us a better perspective of our work. The point-by-point comments were as bellows.

3. Point-by-point response to Comments and Suggestions for Authors

Comments 1: 1. Line 107: Add (Fig. 2C) after “did not.”

Response 1: We added “(Fig.2C) “after did not at Lane 112.

Comments 2: Fig 2C left panel: Explain why the anti-flag band in HIV-1 Gag cells in cells is lower than HTLV-1-Gag.

Response 2: We appreciate the reviewer’s comment. MW of HTLV-1 Gag-precursor protein is ~53kD, while HIV-1 Gag precursor is ~55kD. Therefore, main band of FLAG-tagged protein size HTLV-1 Gag precursor was lower than HIV-1 Gag precursor. We also noticed the additional upper size bands in HTLV-1 Gag precursor, which were always found in the cells, but disappeared in VLPs, indicating the inclusion of some cellular components with HTLV-1 Gag precursor.   

Comments 3: Line 128-130: correct the grammatical error of “Cell lysates and VLPs produced from 293T cells transfected with HTLV-1 Env and control vector, HTLV-1-Gag-FLAG, or HIV-1-Gag-FLAG were analyzed by Western blot using anti-FLAG, LAT-27, and anti-CypA antibodies.

Response 2: We have to apologize our grammatical errors and confusing. The sentence has been changed into “Cell lysates and VLPs produced from 293T cells transfected with control vector, HTLV-1-Gag-FLAG, or HIV-1-Gag-FLAG combined with HTLV-1 Env were analyzed by western blotting using anti-FLAG, LAT-27, and anti-CypA antibodies.”at Line 132-135.

Comments 4: Line 142-147: Explain why N-terminal Fyn was added (containing myristylation and palmitoylation sites) so it can target gag to the membrane. It is mentioned in the discussion, but it should be explained here as well.

Response 4: We added the explanation of Fyn(10) “ single myristylation and a dual palmitoylation signal at Lane 149-150.

Comments 5: Line 142-147: Explain why 6A2T mutation is included in the screen. What is special about this region?

Response 5: We appreciate the reviewer’s comment. The 6A2T is highly basic region mutant, which is important for the plasma membrane targeting of HIV-Gag and budding of virus. This mutant was just used as the negative control of the mutational study. We added the sentence explaining the 6A2T with the reference in Lines 147-149 “highly basic region (HBR) mutant, 6A2T, which loses the plasma membrane binding activity, and does not produce VLPs as previously described [28].”

Comments 6: Line 161: the whole figure 4 is missing in the manuscript pdf, and only two figures are included in the original images of blots/gels so one sub-figure is totally missing.

Response 6: We have to apologize the missing of Figure 4 in the manuscript. We added the Figure 4 in the revised manuscript and added all original images of the Western blot in the supplement.

Comments 7: Line 161-163: please add references of p6 involvement in viral release, and the references of the association of L-domain with VLPs production.

Response 7: We added the reference at Line 172.

Comments 8: Line 172-188: this part is a duplication. Not sure if this is journal or authors’ error.

Response 8: We deleted the duplication part in revised manuscript.

Comments 9:      Line 200: (Fig. 5B) should be (Fig. 5A)

Response 9: We changed Fig.5B to Fig.5A at Line 216.

Comments 10   Figure S2: please improve the quality of this blot. Explain why gp120 band in control cells is lower, and why actin bands are not equal among the 3 lysates.

Response 10: We have to apologize the resolution of Figure S2. As the reviewer suggested, gp120 and β-actin bands in the control cells were little weak compared to knockdown cells. However, HIV-1 Gag expression level of control cells were comparable to knockdown cells, while production level of HIV-1 Gag VLPs in control cells was slightly higher compared to that in knockdown cells. Therefore, it was possible that higher production of VLPs may change the cell condition of control cells.

Comments 11:   Line 208: Add “(Fig. 5B)” at the end of the sentence.

Response 11: We added “(Fig. 5B)” at Line 224.

Comments 12:    Line 263-265: Correct the grammatical error of “p6 domains possess two L-domains, PTAP and YPXL motifs, and are involved in viral release by interaction with TSG101 and ALIX, respectively.”

Response 12: We corrected the grammatical error of this sentence in revised manuscript. “However, p6 in HIV-1 Gag possesses two L-domains, the PTAP and YPXL motifs, which are involved in viral release by interacting with TSG101 and ALIX, respectively” at Lines 282-284.

Comments 13:    Line 265-275: The discussion of VA RNA I is confusing. Where is it located? What is the relationship between VA RNA I with p6, PTAP or YPXL?

Response 13: We are sorry for confusing the reviewer. VA RNA I is adenoviral gene, which hampers dsRNA-mediated inhibition of viral translation. We modified the discussion regarding the role of VA-RNA I in p6 deletion mutants in revised manuscript at Lines 284-293.

Comments 14:   Line 375: Explain the importance of pcDNA-1E-RRE vector.

Response 14: We are sorry for missing the description of pcDNA-1E-RRE, which is a Rev-dependent HTLV-1 Env expression vector. We added description of pcDNA1E-RRE in the methods (4.2 plasmid) in the revised manuscript at Lines 356-357.

Round 2

Reviewer 1 Report

Comments and Suggestions for Authors

The authors have adequately addressed my concerns. I have only two minor points.

1. Fig S3 and S4 seem to be reversed (text references Fig S4 but Fig S3 has the relevant data and vice versa) 

2. In the discussion on lines 323 and 332 the authors talk about the "capsid core" and "viral core" in the context of what they are trying to say, I believe that they mean the "immature Gag lattice" or "Gag lattice". The core of capsid core is usually used to denote the conical structure that forms during maturation. The core is made up of entirely liberated CA proteins and would not, in theory, have any effect on Env incorporation.  

Comments on the Quality of English Language

The English is much improved.

Author Response

We thank the reviewer’s comments. The point-by-point comments are as bellows.

Comment 1. Fig S3 and S4 seem to be reversed (text references Fig S4 but Fig S3 has the relevant data and vice versa) 

Response 1. We apologize for these simple mistakes. They have been corrected in the revised manuscript, at Lines 203, and 218.

Comments 2. In the discussion on lines 323 and 332 the authors talk about the "capsid core" and "viral core" in the context of what they are trying to say, I believe that they mean the "immature Gag lattice" or "Gag lattice". The core of capsid core is usually used to denote the conical structure that forms during maturation. The core is made up of entirely liberated CA proteins and would not, in theory, have any effect on Env incorporation.  

Response 2. We appreciate the reviewer’s comment on this matter. These words have been corrected to “immature gag lattice” at Lines 323, and 332.

We appreciate that the manuscript has been further improved by the reviewer’s valuable comments.

Reviewer 2 Report

Comments and Suggestions for Authors

The authors have responded to the comments sufficiently. 

Author Response

We appreciate that the manuscript has been improved by the reviewer’s valuable comments.